# Could Lavender Farming Go from a Niche Crop to a Suitable Solution for Romanian Small Farms?

Iuliana Vijulie , Ana-Irina Lequeux-Dincă *, Mihaela Preda, Alina Mareci and Elena Matei

Faculty of Geography, University of Bucharest, 1. Blv. Nicolae Balcescu, 010041 Bucharest, Romania; iuliana.vijulie@g.unibuc.ro (I.V.); mihaela.preda@geo.unibuc.ro (M.P.); alina.mareci@unibuc.ro (A.M.); elena.matei@g.unibuc.ro (E.M.)
* Correspondence: ana.dinca@geo.unibuc.ro; Tel.: +40-726-691-272

**Abstract:** Lavender crops have had an impressive continuous development in recent years, being currently a suitable alternative to other traditional crops because they can yield a high profit per hectare. This can be especially useful in Romania, with its high prevalence of subsistence and semi-subsistence farms. This study aims to analyse the issue of small emergent lavender farms in the context of the current Romanian agricultural background, including the framework mechanisms for implementing the Common Agricultural Policy at a national level. The research uses the qualitative survey method to provide broad, synthetic, analytical insights into small lavender farms/businesses in Romania, considering the perspective of the following two target groups: farm owners and civil servants with agricultural expertise. The main results show that both sample groups agree that lavender farms can be successful and satisfactory solutions. Increasing participation in information and training sessions may improve farmers' access to financing mechanisms, but both small farmers and civil servants with agricultural expertise identify a series of problems, mainly regarding the absence of a dedicated market for lavender-based products and a lack of labour force, both essential for maintaining the farming–processing–commercialising chain. The authors also conclude that a more flexible and future harmonisation between Romania's agricultural realities, the Common Agricultural Policy, and the National Rural Development Programme would improve lavender farming's social and economic impact. Follow-up research may envisage more in-depth market analyses for this emerging sector in Romania, facing obvious competition, but which could also benefit from good practice exchanges in the region.

**Keywords:** small farmers; lavender farms; barriers; civil servants with agricultural expertise; perception; tourism; Romania



## 1. Introduction

Lavender is an aromatic plant from the *Lamiaceae* family with approx. 40 species and numerous varieties [1–3] all over the world, and is native to the Mediterranean area, where it has been cultivated since ancient times, starting with the Greeks and the Romans [4–6]. In recent years, it acclimatised to continental climates [7], but its spread continues to be hindered by temperature limitations [5,8]. It is used for its therapeutical properties in curing a series of illnesses [9–12]. Its chemical qualities differ depending on the variety [13,14] and soil [15], and it is also an essential resource for the cosmetics and perfume industry [16,17] and even the food industry [18]. In addition, lavender is valued for its landscaping attributes in urban planning [19,20] and as a tourism attractiveness element [4,21–23], thus playing a significant role in helping small communities by championing investments in rural areas, creating new jobs, and ensuring an increase in residents' income level, ultimately bringing local economic benefits [17,24,25].

Globally, lavender farming has evolved two-fold. At first, Mediterranean states developed it strictly for its economic value in the cosmetics and perfume industry. However,

this crop's applications later diversified, with the focus shifting towards its landscaping value, which was capitalised through tourism. As a result, iconic lavender fields in Provence (France), Isparta (Turkey), Tuscany (Italy), and Croatia became part of rural tourism, promoted through festivals or a new segment—aroma tourism [26].

Presently, the biggest European lavender cultivators are Bulgaria and France [27–31]. Worldwide, the regions or countries renowned for essential oil production are Bulgaria [32,33], France [4], Italy [34], Spain [35,36], Turkey [37,38], Kashmir [39,40], South Africa [41], and regions in Northern Africa [15].

Lavender farming is examined by a whole range of studies belonging to different disciplines that are mainly interested in the lavender oil market [38,39], the ecological sustainability of crops from different points of view [36,39,42] or its economic sustainability for rural and local development [32,37,40,43], including the added value for the tourism sector [44]. Existing research approaches regions with important lavender production results and with tradition in cultivating this aromatic plant [33,37,38,45,46]. To the best of our knowledge, lavender farming in Romania, and especially the stakeholders' perspective on it, has not been approached by scientific studies, despite the growing interest of subsistence small-scale family farms in the economic advantages of this crop.

Lavender farming became important in Romania around 2010, encouraged by the Common Agricultural Policy (CAP) incentive measures and the conditions created by climate change, which constituted a favourable factor for acclimatising aromatic plants typical for Mediterranean Europe. However, compared with countries where lavender has an established tradition and is extensively cultivated, this crop covers mostly small or very small agricultural areas in Romania. This is down to the following two aspects: firstly, it is a new crop introduced relatively late, and secondly, agricultural properties in this country have been transformed and fragmented by historical factors, even more so in recent years. This is proven by the fact that several studies dedicated to small-scale farming and its typology in Europe [47] or particularly to its efficiency and sustainability for Central and Eastern Europe (CEE) that have undergone critical socio-economic transformations during the post-communist period [48], inevitably also included Romania. In fact, according to the National Institute of Statistics, out of the total number of agricultural holdings in Romania, those smaller than 1 ha make up 53%, and those between 1 and 5 ha make up 38.6%, but only cover 28.7% of the utilised agricultural surface of the country [49]. This study aimed to evaluate, from an empirical perspective, small-scale lavender farming as a suitable economic and environmental solution in the light of the European Union's Common Agricultural Policy. In this respect, the perspectives of both farmers and civil servants with agricultural expertise were surveyed through semi-structured interviews.

The research gap in this specific type of analysis stems from the fact that studies on lavender farming in Romania have focused on technical information about planting/cultivation. Most of the published works are, in fact, reports detailing how lavender should be grown [50–52], climate and soil requirements [50,51,53], details on lavender varieties [54,55], maintenance works [52,53,56], production [54], harvesting [52,54], and on the manufacturing and use of lavender essential oil [53]. However, to date, there are no articles that qualitatively investigate the perspective of farmers and decision-makers regarding this niche crop, despite the growing interest of subsistence and semi-subsistence family farms in the economic benefits of lavender farming. Thus, the economic and social problems that lavender farmers face, their perception of the success of their business, and the effects of this crop on local communities have not yet been addressed by specialised studies.

The layout of the manuscript mirrors its exploratory approach, which sought to capture the complexity of lavender farming on small plots in the local agricultural landscape as an alternative to other traditional crops because they can produce a considerable profit per hectare.

The article was structured into the following five parts: introduction, methodology, results, discussions, and limitations of the study, ending with the chapter on conclusions. Thus, the study opens with an introductory chapter dedicated to the literature review,

which starts with a general view of the topic and then focuses on a particular aspect of its relationship with the study area. This is followed by a presentation of the geographical and environmental considerations of lavender culture in Romania as well as EU agricultural policies that support its local development through a series of measures and sub-measures to support small farmers with the help of specialised institutions in the country.

The next chapter describes the research methodology, which used the qualitative survey method to provide broad analytical perspectives on small lavender farms in Romania, taking into account the views of the following two target groups: small lavender farm owners and civil servants with agricultural expertise. The article continues with the main results and discussions, including the study's limitations, followed by the conclusions of the research.

### 1.1. Background of Lavender Farming in Romania

1.1.1. Geographic and Environmental Considerations

Out of all aromatic plants, lavender is the most well known in Romania, even if it is a relatively recently introduced crop. The first bushes were recorded to have been planted in the 1950s, around Bucharest, probably by Bulgarian gardeners [56]. Afterwards, this crop extended south and west. In 1990, over 3000 ha were planted with lavender. However, between 1990 and 2005, this coverage was drastically cut to approx. 420 ha, with most farms located in southern Romania [57–59].

The year 2010 saw a gradual increase in lavender farming in almost all country regions, but the crop was mainly cultivated on small farms. According to recent data from the Agency for Payments and Intervention in Agriculture—APIA (Agenţia de Plăţi şi Intervenţii pentru Agricultură—APIA), lavender farms with an area of between 0.03 and 2 ha make up 86.18% of the total farm number and are supported by very small households practising subsistence farming. The next type of plots cultivated with lavender is farms between 2 and 5 ha, which constitute 10.9% of all farms number, being small semi-subsistence households. In contrast, the most extensive farms, covering between 5 and 25 ha, are owned by companies specialising in lavender farming but only make up 2.92% of their total number (Table 1) [60].

**Table 1.** Lavender farms in Romania.

| No. | Area Cultivated with Lavender per Locality (ha) | Total Coverage (%) | No. of Localities with Lavender Farms per Category |
|-----|---------------------------------------------------|--------------------|-----------------------------------------------------|
| 1 | <2 ha | 86.18% | 237 |
| 2 | 2 ha–5 ha | 10.9% | 30 |
| 3 | > 5 ha | 2.92% | 8 |

Source: APIA data, 2019 [60].

Lavender is cultivated in many areas of Romania with a high insolation degree due to the plants' ecological adaptive and drought-resistant capabilities and because it has no particular demands on types of soil—though it prefers well-drained ones [53].

Most Romanian lavender farms are centred on very small and small agricultural plots (<5 ha), which make out 97.08% of the total farms' number. Their national distribution is unequal, but the most crucial region for lavender farming is the Transylvania Macroregion (MR1), with its high concentration of farms accounting for 42.69% of their total number. The second most important macroregion is Moldova-Dobrogea—MR2 (26.23%), followed by Banat-Oltenia—MR4 (16.85%) and Muntenia—MR3 (14.23%).

Conversely, all larger farms (5–25 ha) are located in just a few localities (e.g., Mihail Kogălniceanu, Corbii Mari, Tecuci, Brăganul, Poiana, Nicolae Bălcescu, Simeria, and Pischia) in southern Romania (in Moldova-Dobrogea—MR2, Muntenia—MR3, and Banat-Oltenia—MR4 Macroregions). The latter area's territorial distribution and the associated sizeable lavender production farms are explained by the optimal pedoclimatic conditions

in the southern parts of the country that resemble lavender' Mediterranean regions of origin (Figure 1).

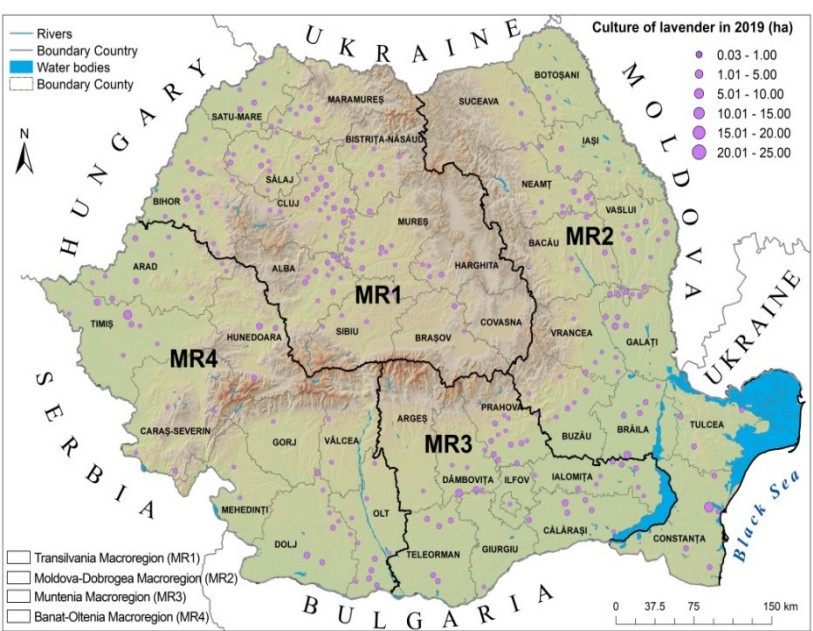

**Figure 1.** Areas covered with lavender farms per locality in Romania (2019). Source: APIA data, 2019.

### 1.1.2. Farming and Lavender Farming in Romania

Lavender farming in Romania is mainly limited to small farms, which is a dominant feature of the Romanian agricultural landscape. Subject to profound transformations induced by the transition from a socialist to an open market economic system, family farming and small-scale agriculture are dominant in the CEE countries, among which Romania distinguishes itself with the largest number of small farms [48].

These types of farms are excessively numerous in our country and represent 33% of the total European Union's (EU) farms of approx. 9.9 million [61,62]. Called family farms in the EU, in Romania they could be defined as both subsistence farms (smaller than 2 ha with a less than 2000 euro/year standard economic value) as well as semi-subsistence farms (between 2 and 5 ha with an economic value of 8000 euro/year) [63]. Subsistence farms mainly produce for the household's consumption and are managed and worked by family members, often the elderly. As far as 50% or sometimes more family members manage semi-subsistence farms to ensure household consumption and commercialise a small part of their agricultural production [62,64,65]. According to the international/European size hierarchy, these farms can be categorised into very small and small farms [66]. Many of them face obstacles such as poverty and a dire lack of labour, leading to reduced economic competitiveness and productivity [67]. Reasons for this include farm fragmentation (i.e., nowadays, a Romanian agricultural holding is formed of multiple small plots) [68], existing demographics, the commercial and technological context, and severe depopulation of rural areas. Moreover, in many cases, they might disappear due to larger farms absorbing them or because they are straightforwardly abandoned [69].

The multitude of small farms in the eastern EU—a phenomenon called "rural subsistence" by the specialised literature [70]—is a challenge for rural development. Moreover, it will continue to be a challenge as their presence will persist due to historical, political, and financial factors and the mentality of former communist states [71]. Many Romanian farmers rejected the idea of associations after the 1990 land re-allotment government measures [72,73], as they still lament the communist collectivisation process (1949–1962) when they were forced to surrender their properties to the Agricultural Production Cooperatives [74–76]. Furthermore, the Romanian landowners who nowadays form associations usually receive insignificant financial remuneration, with most economic benefits going

towards those who organised the association [77]. As such, landowners of small farms further lose motivation to associate, even despite successful examples in their vicinity (e.g., the Moldova Lavender Association, which recently received member status in the European Federation of Essential Oils—EFEO) [78]. The Moldova Lavender Association showcases an excellent example of how niche crops such as lavender can be "profitable". They can be a lucrative option that would allow farmers or landowners of subsistence or semi-subsistence farms to continue their activities in their current form when faced with the emergence in their vicinity of more extensive agricultural holdings [79–81].

Even though they face some obstacles and limitations, lavender farms are expanding in Romania. Many specialised producers and firms are creating new plantations, processing and/or exporting raw products or essential oils [59]. Furthermore, the following characteristics of lavender farms facilitate their success: the plant requires minimal work and low maintenance costs, and the finite products obtained through only primary processing bring significant added value.

### 1.1.3. EU Agricultural Policies

After Romania's EU accession, the Common Agricultural Policy (CAP) implementation has supported rural development and sustainable agriculture [82]. Created in 1962, CAP is dynamic, adaptable, constructive, and committed to answering the UN Sustainable Development Goals (SDG) 2030. Although CAP is preparing for the 2023–2027 exercise, it continues the actions of the 2014–2020 exercise regulations by following its three domains as follows: direct support [83], marketing measures [84], and rural development [85].

CAP values small farms considering they generate the following: picturesque rural landscapes, biodiversity, local products, and they maintain traditions and ethnocultural specificities. Moreover, small farms substantially increase rural areas' attractiveness for business, rural residency, or leisure and relaxation activities for tourists [86].

In transposing CAP at a national level, the Romanian Ministry of Agriculture and Rural Development (MARD) had to support farmers by activating a series of instruments to consolidate agricultural competitiveness using programmes and measures that energise agricultural activities through subsidies and loans or guarantees. The Ministry implements European policies via its 42 county agricultural directorates (CAD), 8 regional centres, and 42 county offices of the Agency for Rural Financing and Investments—ARFI (Agenţia pentru Finanţarea Investiţiilor Rurale—AFIR), and the 42 county centres of the Agency for Payments and Intervention in Agriculture—APIA (Agenţia de Plăţi şi Intervenţii pentru Agricultură—APIA) [87–90].

Among the financial measures (M) and sub-measures (sM) that aim to support farms (including lavender) as well as other agricultural activities in Romania within the National Rural Development Programme (PNDR) implemented through the ARFI relevant for this study are *sM 6.1. Setting up young farmers* (112 measure from the previous exercise); *sM 4.1. Investments in agricultural holdings; sM 4.2. Component for agricultural processing; sM 17.1. Mutual funds: helping farmers with insurance premiums, covering agricultural risks, economic losses due to climate hazards, plant diseases, pest infestation, or environmental accidents* [86].

Within the PNDR 2014–2020, APIA finances *M 10. Environment and climate* and *M 11. Ecological agriculture,* and farmers can apply to both these ongoing programmes for yearly subsidies [91–93]. Similar to any other agricultural entrepreneur, lavender farmers can, if eligible, receive financial support from multiple measures and sub-measures. For example, farmers who meet the eligibility criteria can apply to *M 10.4 Agro-environment—green crops*—and receive a yearly subsidy of 130 euro/ha [94]. Among the eligibility criteria, the most important one is that the minimum surface of the cultivated plot is 0.3 ha. For farms smaller than 1 ha, this is the main impediment to applying for APIA subsidies (Table 2).

**Table 2.** Main types of lavender farming in Romania and associated elements.

| Type of Farm | Very Small Farms/ Subsistence Farms | Small Farms/ Semi-Subsistence Farms | Larger Farms |
| --- | --- | --- | --- |
| Dimension | <2 ha | 2–5 ha | >5 ha |
| Type of owner | Farmer | Farmer/Company | Company |
| Access to EU financing mechanisms | Conditioned | √ | √ |
| Business financing | Private savings/Bank loan/SF (structural funds) | Private savings/Bank loan/SF (structural funds) | Private savings/Bank loan/SF (structural funds) |
| Degree of using /accessing SF (structural funds) | Very weak | Weak | Moderate/good |
| Labour force structure | Mainly family members | Family members + employees | Mainly employees |
| Distribution | X | √ | √ |
| Production's orientation | Other purposes + oil | Oil + other purposes | Oil + other purposes |
| Other associated activities | √ | √ | √ |

Source: Authors composition and APIA data, 2019.

The farmers who are certified as practising ecological agriculture can benefit from M 11., and receive support from the more suitable of the two sub-measures available, either *sM 11.1. Support for converting to ecological agriculture practices and methods* for which they receive 365 euro/ha/year, or *sM 11.2. Support for maintaining ecological agriculture practices and methods,* for which they receive 350 euro/ha/year [88]. In addition to these financial mechanisms, APIA offers compensatory payments to lavender farmers in the unique payment-per-plot land scheme (SAPS) with a total of 80 euro/ha, and a national transition support (ANT 1) regardless of product with a total of 20 euro/ha, which means that a farmer can theoretically receive subsidies of up to 600 euros per year because the M and sM payments are cumulative for the same plot if the farmers are deemed eligible [94].

One important aspect worth mentioning is that despite this crop's opportunities in terms of agricultural profitability, there are no political measures or stipulations particularly adapted for marketing lavender products. On the contrary, the market integration of agricultural holdings is generally emphasised as a significant problem for small-scale farms in Romania [95]. In terms of creating market access for food products from small farms in Romania, studies have shown few successful attempts. These products contribute to a large extent to the 'informal food exchanges,' partially leading to the so-called 'unseen food' or 'self-provisioning' products [96]. As a cautionary tale, when national measures to support various crops (e.g., the Tomata Programme, the Garlic Programme) are implemented, farmers cannot capitalise on the excess of their yearly production because many of them lack training and essential experience for basic business tools and operations. The excess production cannot be absorbed by the local consumer market or supermarket supply chains, as these regularly appeal to imported products to the detriment of local farmers [97]. Moreover, the same distribution problems are even more prevalent in the case of small or very small farms (Table 2), which represent the target of the current study and, especially for lavender as a new type of crop, need a different market than the one for food products.

In this context, it is even more surprising that those who capitalised first on the opportunities brought forward by this niche crop were precisely the owners of small land plots (i.e., confronted most often with self-production agricultural activities and income

insecurity) and not political actors or large corporate entities. Over time, business actors such as banks started offering designated loans for profitable agricultural endeavours. Transilvania Bank is worth mentioning, as it created a series of financing measures dedicated to lavender farming in Romania (i.e., it publishes and promotes a Complete Guide for Lavender Farming in Romania) [98] and even defined in business terms the "small farmer" and encouraged start-up lavender businesses with dedicated personal loans [99].

## 2. Data and Methodology

In order to meet the aim of the study, of analysing the situation of the small emergent lavender farms within the present-day agricultural Romanian background, this research was based on qualitative surveys to provide broad analytical insights into small lavender farms from the perspective of the two target groups.

The owners of the small and very small lavender farms were interviewed first to obtain an overview of their perception of lavender-based businesses following a series of variables relevant in the particular regional context of Eastern Europe.

This study focused on two types of farms defined by the EU and adapted to the Romanian context according to multiple factors (e.g., the size of cultivated area, associated economic benefits, etc.) (Table 2). The first refers to subsistence (very small) farms and the second to semi-subsistence (small) farms. These two types were considered because of their high number and socio-economic importance in Romania. Farms smaller than 2 ha and those between 2 and 5 ha make up more than 97.08% of the total number of lavender farms (Table 1). This study focused on these specific types of farms because of the complexity of problems their owners face and the overall acute need to stimulate agricultural entrepreneurship (Table 2). In order to simplify their description, the owners of small and very small farms interviewed by the authors will be referred to as "small farmers".

The second set of interviews was conducted with civil servants with agricultural expertise because their opinions on the present situation are relevant to the future perspectives of developing lavender farms in Romania in the framework of applying CAP.

### 2.1. Data Source and Methods

The secondary data were mined from the Romanian National Institute of Statistics and APIA (total area cultivated with lavender in 2019, number of farms, farm surface, data on CAP and MARD operational structures) for each macroregion of the country (MR1, MR2, MR3, MR4). Data from APIA regarding the location of lavender farms in the country were mapped using the proportional symbols cartographic method in ArcGIS Pro 2.8 (ESRI, Redlands, CA). On this map, Romania's four macroregions were identified to more clearly represent the level of development of lavender farming according to its territorial spread (Figure 1).

Primary data in our research came from two in-depth qualitative surveys interviewing both small farmers and civil servants with agricultural expertise. The latter category included people working in the following local agricultural branches of MADR: CAD, ARFI, and APIA, all of which have been responsible for managing CAP programmes.

The small farmers were selected randomly with the help of Romanian lavender farmers' associations from the country's four macroregions, who distributed information about our research intentions on specialised social media groups (Facebook, WhatsApp). Afterwards, associations facilitated authors to contact the participants directly by phone to collect their answers in the first phase of February–March 2020.

Authors used the e-mails from official websites to contact civil servants with agricultural expertise from the three prominent institutions (CAD, ARFI, and APIA) located in each of the 41 counties and the capital city of Bucharest to establish details about the most convenient interviewing channel. Bucharest and two counties (Ilfov and Covasna) were afterwards excluded from the sampling because they included no lavender farms. In this way, we conducted asynchronous interviews by e-mail between December 2019 and March 2020, primarily because of Romania's pandemic-related restrictions. Secondly, this was a

more manageable (less time and money consuming) way also used in other studies [100], which was maintained in the second interviewing phase.

The second interviewing phase took place in March 2022 to achieve an optimal sampling size in relation to the study's analytic purposes and methods. This was performed by e-mail as a more accessible and preferred communication channel in the case of civil servants with agriculture expertise and by both phone and face-to-face interviews in the case of the small farmers. The latter interviews were possible in the context of pandemic restrictions relaxation. They took place at the Lavender Fair "Lavandişor" in Bucharest, one of the most significant events where small lavender farmers from all the country's regions converge annually. Both surveys were performed according to the academic requirements and EU legislation on managing personal data [101]. Each interview started with a presentation section detailing the aim of the study, authors' affiliation, details on the personal data confidentiality, and participants' agreement.

### 2.2. Methodological Tools and Sample Size

Surveys and interviewing methods targeting stakeholders represent a broadly utilised technique for obtaining data in studies that approach the topic of small farmers [37], their business practices, and market participation [102,103], especially when considering recently introduced crops, for which external governmental stimuli and start-up financial loans are vital [104,105]. In addition, interviewing methods are successfully applied in exploratory studies that provide empirical insights into the small farm household sector [106].

We aimed to fill a research gap about the business experience and perspectives of small lavender farmers in Romania, which has not previously been considered as far as we know from the existing scientific literature. This study had an exploratory approach that aimed to capture the complexity of this emerging sector in the autochthonous agricultural landscape.

In this respect, the semi-structured qualitative survey, containing mainly open questions and some Likert scale questions, was the most appropriate method to identify the relevant issues, dimensions, and characteristics of our topic of interest [107] and corresponded to the rationale of our approach. This method allowed us first to obtain raw data from interview transcripts and then cover the complexity of the topic while avoiding the limitations imposed by a pre-structured or deductive type of qualitative survey [107]. In a second stage, data were coded and transformed mainly into nominal variables in SPSS v.23 (IBM), allowing us analytic generalisations and certain empirical quantitative approaches necessary for "systematic comparisons" [108], p. 7. Jansen [107] confirms the general practise of inserting quantitative approaches (e.g., cross-tabulation of data) into qualitative research in the current "paradigmatic situation," which increased permissiveness toward mixing methods [109], p. 2. Despite several quantitative techniques utilised to analyse nominal data in SPSS (e.g., cross tabs and Spearman coefficient—particularly suitable for nominal data), the current research approach was meant to explore the topic and depict the diversity of the sample target population and interviewed groups rather than emphasise "numerical distribution" [107], p. 3. Quotations from both small farmers and civil servants consistently complement the analyses to explain and further validate the research hypothesis. In order to keep the anonymity of the respondents, the quotes were coded "F" for farmers and "CsA" for civil servants with agriculture expertise. In addition, the code also includes the names of the county and macroregion of each respondent.

Besides the possibility to emphasise the opinion diversity of a sample group for a certain phenomenon—in our case of the small-scale lavender farming business in Romania—the coding process allowed us to adjust the sample to an optimal satisfactorily size through data saturation techniques. The sample dimension in qualitative research is debatable and varies among scientists [110–113].

Researchers consider data saturation, a relevant parameter, as achieved, thus allowing a sufficient confidence level in survey results when no new answers are offered. Moreover,

particularly in qualitative research, "the goal of sampling is to collect data that either further develop or challenge existent hypotheses", as this study also attempted [114], p. 70.

Taking into consideration all the above, as well as the territorial extension of this agricultural crop in Romania and the scientific paradigm of our research [112], we adjusted our sample size for small-scale lavender farmers in Romania to a satisfactory level of 162 respondents. Officially, only 267 small and very small lavender farms are registered by the APIA statistics (Table 1) as they are currently receiving subsidies for this crop through this institution. Their number is, in fact, higher, but official data in this respect are missing creating a limitation for the present study.

In the case of civil servants with agricultural expertise, the sample numbered 47 valid answers that were considered to represent optimal institutional and territorial coverage. These 47 respondents came from all the following above-mentioned agricultural institutions: 39 respondents work within CAD (1 answer for each of the 39 counties—NUTS3—with lavender farms); 4 respondents work within AFIR; 4 respondents work within APIA (1 answer from each institution on a macroregion scale—NUTS1). We chose to focus on CAD employees because this institution is the one responsible for organising promotional and training programmes for farmers regarding the PAC measures and sub-measures mentioned in sub-chapter 1.1.3 EU agricultural policies (M 10., M 11., sM 6.1., sM 4.1., sM 4.2., sM 17.1.). AFIR and APIA are tasked with European fund absorption by implementing EU-funded projects for the first and granting subsidies in the case of the latter.

*2.3. Research Hypotheses*

The surveys for both target groups included addressing the three qualitative research hypotheses, namely, H1: Small farmers' participation in training programmes leads to their accessing structural funds; H2: The social and economic impacts of lavender farms on Romanian rural communities are low; H3: Lavender farming has mostly beneficial ecological effects.

**Hypothesis 1 (H1).** *Small farmers' participation in training programmes leads to their accessing structural funds.*

This is an intuitive hypothesis, as once a beneficiary participates in a training programme for accessing European funding, his or her ability to do so would improve. In this regard, the Rural National Development Programmes for 2015–2020 propose, through Pillar II, different activities such as those organised by authorised institutions oriented towards financial instruments. Other types of proposed activities are more applicable as follows: training by farmers for farmers, farmers' working groups where they can share know-how with other countries and vice versa, learning to use capital to provide equipment for certain kinds of competencies (e.g., processing the raw produce in-situ), and demonstration events [115].

Training programmes that target helping farmers access structural funds are a series of tools aimed at increasing the absorption rate of European funding. However, the implementation of these programmes differs among EU member countries. The participation rate of farmers in the training programmes and the degree to which they access European funds are closely related to the organisation and functioning of the institutions responsible for carrying out the training programmes (CAD, ARFI, and APIA). This can sometimes represent failure factors due to bureaucracy, lack of communication between public authorities at all levels, or sometimes expertise and knowledge gaps of some of the employees [116].

The willingness of farmers to participate in training programmes dedicated to financing and establishing new or agri-environmental crops does not always guarantee the success of the entrepreneurial initiative [117]. The reluctance regarding training programmes is determined by their theoretical specificity, the limited specialised field the farmer is interested in (vegetable grower, apple-grower, etc.), as well as the (im)possibility of learning about successful experiences by directly exchanging know-how with other

farmers [118]. Participation in training programmes has had a positive impact on accessing different funding axes and developing initiatives in agriculture, with interest in participation being higher among young people than among other age groups [119]. EU statistics show that in 2016, about 19% of young farmers had received full agricultural training compared to 2.6% of farmers over 65 years of age, which led to an increase in the number of investments in small and medium farms set up by young people (i.e., 27.5% of EU farms) [120,121]. However, we can say that young farmers still face significant difficulties in obtaining funding, given the large investments and the low yield of a farm in its infancy/start-up phase [122].

**Hypothesis 2 (H2)**. *The social and economic impacts of lavender farms on Romanian rural communities are low.*

This hypothesis was constructed to mirror the reality of Romanian agriculture, namely, the predominance of small plots, which will probably continue to manifest due to farmers' hesitance to associate. Hence the agriculture practised on a small scale would have a negligible impact. From an economic point of view, introducing lavender farming into rural areas is advantageous considering their high investment return. However, although lavender has the potential to increase their income, farmers' profits can be small if production planning does not correlate with market consumption [123]. Lavender farms' profitability must also weigh the general costs of labour wages in the region, renting agricultural equipment, and, last but not least, reinvested capital. This being said, low-profit margins pressure farmers to keep their traditional activities—animal husbandry and cultivating traditional crops—as a primary source of income. Their profits will also decrease if they do not benefit from any form of subsidies or if they are not part of a larger agricultural organisation that can develop a high capacity to process, promote, and commercialise their products [37].

Farmers who diversify their activities and include other crops besides the traditional ones also benefit from governmental support through specific financing mechanisms [124]. From a social point of view, lavender farming can revitalise an area by creating new jobs due to a diversification of the local economic activities, leading to increased incomes and the involvement of more community members in a shared venture, thus strengthening social cohesion. Apart from generating income from raw material processing, lavender fields also encourage tourism and trade entrepreneurship, which in cases such as France have proven to be very profitable [125]. Other possible positive social effects include decreasing the risk of demographic ageing by motivating young people to remain in rural areas as the new jobs are more suitable to their preferences and training (product designers, managers, tourism guides, promoters, internet content creators, etc.) [22,126].

Lavender tourism success rate directly depends on the region's annual tourism capacity, with its development becoming viable if it complements an already existing rural tourism market. Rural areas already known as tourism destinations that have introduced lavender fields into their portfolio were able to benefit from diversifying the range of their tourism offer with products such as breakfasts or tours in lavender gardens, lavender-scented village scenery combined with unique nature walks, photo shootings, as well as from commercialising lavender-based products (oil, scented water, honey, soap, jam, potpourri, pillows) [17].

In other words, tourism can contribute to the future increase in the living standards of rural communities' members if farmers take into account and incorporate the best examples of good practices in countries such as France, Turkey, Croatia, etc. on the tourist exploitation of lavender crops [127–129].

**Hypothesis 3 (H3).** *Lavender farming has mostly beneficial ecological effects.*

The hypothesis started from the results of previous studies that showed that lavender, in general, has had a positive effect on the environment as its farming does not cause significant damage to the quality of the air, water, and soil [130]. At the same time,

lavender is a source of pollen and nectar for bees and is important for the local biodiversity, environmental balance, and the functioning of its ecosystem [131].

Examples of positive effects and reduced environmental impact of lavender farming which were identified by the specialised literature are linked to beekeeping and biodiversity—bio lavender crops, in particular, contribute to increased biodiversity as the plant is a magnet for many species of insects and birds (butterflies, bees, pheasants, or families of kestrels—*Falco tinnunculus*), which can find lavender fields as a perfect habitat [132–134]; natural insecticide—aromatic plants such as lavender could be an adequate alternative to chemical insecticides (they could be a healthier option to the chemical substances that plants are pulverised in order to eliminate pests) that would also target the share of the population that prefers to consume bioproducts (foods that have not been treated or treated with natural-based insecticides or pesticides) [135–139]. These fields also create a pleasing aesthetic landscape [127,140], with the added bonus of strong fragrances released by volatile oils, allowing for a natural aromatherapy session in the middle of lavender fields [36,137].

Authors such as Fayet [141] show in their study that landowners perceive EU CAP policy as a powerful tool that can strongly influence farmers to stop land abandonment and stimulate opportunities for redevelopment of agricultural land. Lavender is not a very demanding shrub; it adapts easily to degraded, eroded, or abandoned lands and positively influences the environment [53].

### 2.4. Survey Content and Sample Characteristics

The surveys for both target groups were semi-structured and started with questions referring to socio-demographic data (Tables 3 and 4).

**Table 3.** Socio-demographic profiles of farms owners (small farmers).

| Variable | Item | % | Variable | Item | % |
|---|---|---|---|---|---|
| Gender | Male | 54.3 | Occupation | Labourers (plumbers, tractor drivers, car mechanics, couriers) | 46.3 |
| | Female | 45.7 | | Farmers | 19.8 |
| Age groups M = 43.98 | 25–34 | 21.6 | | Legislators and higher officials (public administration officials or lawyers) | 6.7 |
| | 35–44 | 32.1 | | Other professionals (engineers, economists, IT technicians, doctors) | 27.2 |
| | 45–54 | 30.3 | | | |
| | 55–64 | 16.0 | | | |
| Marital status | Married | 87.7 | | | |
| | Not married | 12.3 | | | |
| Education | Primary | 3.7 | Source of funds for starting the farm | Private savings | 80.3 |
| | Highschool | 40.7 | | Private savings/Bank loan and SF (Structural Funds) | 12.9 |
| | University or higher | 55.6 | | | |
| Agricultural education | No | 71.6 | | Private savings and SF (Structural Funds) | 6.8 |
| | Yes | 28.4 | | | |
| Participated in a training activity related to CAP/PNDR | No | 71.6 | Regions | MR 1 (Transylvania) | 34.6 |
| | M 11. | 17.3 | | MR 2 (Moldova-Dobrogea) | 30.3 |
| | sM 6.1 (former M 112) | 10.5 | | MR 3 (Muntenia) | 14.2 |
| | sM4.1 | 0.6 | | MR 4 (Banat-Olt) | 20.9 |

**Table 4.** Socio-demographic profiles of civil servants with agricultural expertise.

| Variables | Item | Frequency (%) | Variable | Item | Frequency (%) |
|---|---|---|---|---|---|
| Gender | Male | 44.7 | | MAKIS-2009 (Modernizing Agricultural Knowledge and Information Systems) | 14.9 |
| | Female | 55.3 | Last attended programme | | |
| Education | Highschool | 8.5 | | No attendance | 19.1 |
| | University or higher | 91.5 | | CAP 2014–2020 | 38.3 |
| Institution | CAD | 83.0 | | PNDR 2014–2020 | 23.4 |
| | ARFI | 8.5 | | ARFI training activities | 4.3 |
| | APIA | 8.5 | | MR 1 (Transylvania) | 25.5 |
| Age groups | 25–34 | 8.5 | NUTS | MR 2 (Moldova-Dobrogea) | 34.1 |
| | 35–44 | 17.0 | | MR 3 (Muntenia) | 19.1 |
| | 45–54 | 25.6 | | MR 4 (Banat-Olt) | 21.3 |
| | 55–64 | 48.9 | | | |

For the first hypothesis, questions for farmers focused on their business experience, financing outlets used by farmers to set up their lavender business (private savings, structural funds (SF), bank loans), and their attendance at information dissemination or training programmes organised by the authorities about various CAP measures and sub-measures.

Open-ended questions meant to present both the economic and social impact of lavender farming were further coded into *family subsistence and tourism development*, respectively. *New jobs* and *new business trends* were formulated to test the second hypothesis.

The Likert Scale (1—very dissatisfied, 5—very satisfied) was used to measure farmers' perception of their satisfaction level towards this particular business. Closed questions that were afterwards coded into ordinal variables addressed business development (e.g., lavender farming, processing, selling) and business perspectives.

Open-ended questions regarding the respondents' opinions on the ecological effects of lavender farming were included in order to verify the third hypothesis.

Questions for civil servants with agriculture expertise mainly followed the same structure, dictated by the three main research hypotheses, but some questions were adapted for this group of respondents (e.g., if their organisation had implemented farmers' training programmes under CAP measures; Likert scale questions on their perception of the success of lavender business in Romania, which are explicitly the success factors for such a business; to name both the positive and negative ecological environmental effects of these cultures; to exemplify possible agricultural measures necessary in farming this niche crop).

Both questionnaires end with two questions addressing barriers and solutions for developing lavender business farming in Romania (see Supplementary Material).

## 3. Results

### 3.1. Farmers Attendance to Training Programmes on CAP/PNDR Measures

Of the farmers that attended the training and information programmes regarding structural funds, more than half participated in meetings on M 11. (Ecological agriculture) and more than a third on sM 6.1. (Setting up new farmers).

Crosstabulation between the type of financing and training and information programme attendance shows that 18.2% of respondents who accessed structural funds had participated in training and information programmes organised on CAP measures and sub-measures. The chi-square test result for this table rejects the null hypothesis regarding the association between these two variables. The results for the ($X^2(2) = 7.735$, $p = 0.021$) test show there is enough evidence to suggest an association between the type of financing and programme attendance as $p$ is below the considered significance level of (a = 0.05).

The association between education level and programme attendance is not statistically significant, and the crosstabulation percentages do not prove that the attendance level increases with their education level. Therefore, it seems that participation in a training or

information dissemination programme is mainly influenced by their personal motivation to have a successful lavender business and not by their education level.

This aspect is also confirmed by the Spearman correlation coefficient, which showed a positive statistical relationship between the level of attendance at these programmes and the satisfaction level of small farmers regarding the development of their lavender business. The two-tailed *p*-value rejects the null hypothesis, and the correlation coefficient has a value of 0.325 (Table 5).

**Table 5.** Spearman correlation coefficient between programme attendance and satisfaction level.

|  |  |  | Programme Attendance | Satisfaction |
|---|---|---|---|---|
| **Spearman's rho** | Programme attendance | Correlation Coefficient | 1.000 | 0.325 ** |
|  |  | Sig. (2-tailed) | . | 0.000 |
|  |  | N | 162 | 162 |
|  | Satisfaction | Correlation Coefficient | 0.325 ** | 1.000 |
|  |  | Sig. (2-tailed) | 0.000 | . |
|  |  | N | 162 | 162 |

** Correlation is significant at the 0.01 level (2-tailed).

The civil servants with agricultural expertise unanimously declared that their institutions organised CAP-related information dissemination and training programmes. Furthermore, approximately two out of five respondents confirmed knowing small farmers who had applied for structural funds as a result of attending such programmes, further confirming our hypothesis. A significant percentage of 21.27% of this target group declared, however, that they did not know or had no information regarding either the attendance level of small farmers in training programmes or the degree to which small farmers' attendance had led to them accessing structural funds. This proves the flawed communication between civil servants and small farmers or the lack of frequent monitoring of the results obtained after attending training and information programmes. These kinds of issues were also brought forward by studies in other countries [116]. The lack of awareness from the civil servants with agricultural expertise could result from personal factors, expertise, or specific knowledge of certain employees.

The degree to which authorities (e.g., CAD, AFIR, APIA) organise and follow up on these events influences the level of farmers' participation in the training programmes and their applying for and obtaining European funds. However, many times, research studies have noted the critical failure of this relationship due to bureaucracy, crucial delays, and lack of communication among public authorities at national, regional, and local levels, and the low level of expertise and knowledge of particular employees or authorities, especially at regional and local levels [116].

The low rate of participation in these programmes could be a factor that leads to low information levels and, thus, to farmers not accessing structural funds. Only a small part of the interviewed small farmers financed their business through structural funds but combined with their private savings (6.79%).

*3.2. The Social and Economic Impacts of Small-Scale Lavender Farming on Romanian Rural Communities*

The economic impact of lavender farms was coded as a source for family subsistence and the potential for tourism development.

Most interviewed small farmers (63.58%) declared that their lavender business is primarily a source of income for their family, showing the low economic impact of these small and very small lavender farms, known in Romania as semi-subsistence and subsistence farms.

The crosstabulation between the economic impact of lavender farming and the satisfaction level regarding their lavender business shows that most of both categories (those who identify the economic impact of lavender farming as "family subsistence" or "tourism development") are partly satisfied or satisfied (more than 30% of each category) (Figure 2).

This proves that the presence of lavender farms generates income primarily for family members who are engaged in the day-to-day agricultural work, thus having a social impact associated with the economic impact, which is explicitly connected but unfortunately limited to each lavender farm business.

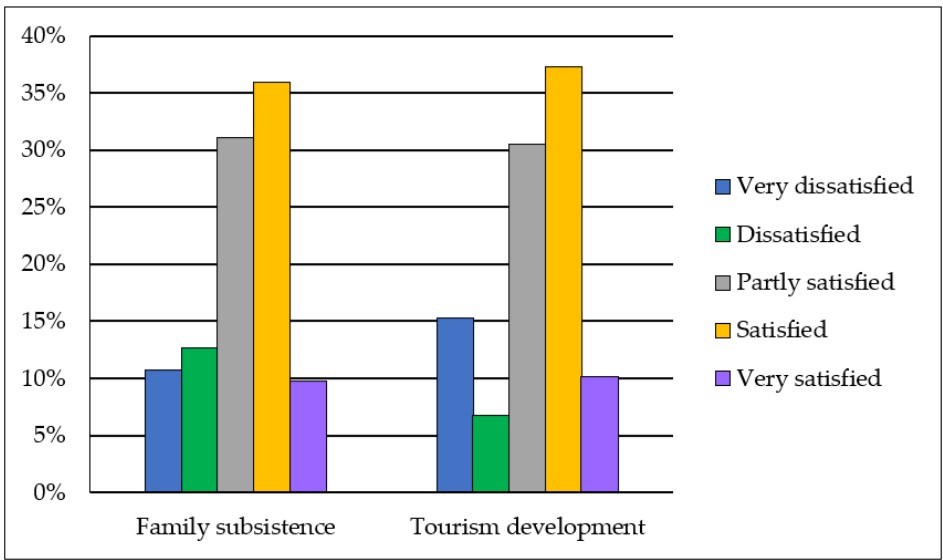

**Figure 2.** Crosstabulation between economic impact and business satisfaction level.

Lavender farms diversify economic activities at the local community level, and their importance was perceived differently depending on small farmers' education level. Thus, small farmers with primary and high school education levels overwhelmingly declared that the farms ensure their subsistence-level incomes. In contrast, those with university or higher levels of education identified the potential for entrepreneurial development of lavender farms in domains such as tourism (Table 6).

**Table 6.** Crosstabulation between economic impact and education level.

| | Economic Impact | | Total |
|---|---|---|---|
| **Education Level** | **Family Subsistence** | **Tourism Development** | |
| Primary | 83.30% | 16.70% | 100.00% |
| Highschool | 80.30% | 19.70% | 100.00% |
| University or higher | 50.00% | 50.00% | 100.00% |
| Total | 63.60% | 36.40% | 100.00% |

The weight of answers naming tourism development, in other words, a diversification of the economic profile of lavender businesses, represents a superior level of their economic impact and is influenced by small farmers' education level and occupation. This is proven by the Spearman correlation coefficient, with a value of 0.313 showing a significant statistical association between education level and economic impact (Table 7).

**Table 7.** Spearman's rank correlation coefficient between farmers' education level and farms' economic impact.

| | | | Education Level | Economic Impact |
|---|---|---|---|---|
| **Spearman's rho** | Education level | Correlation Coefficient | 1.000 | 0.313 ** |
| | | Sig. (2-tailed) | . | 0.000 |
| | | N | 162 | 162 |
| | Economic impact | Correlation Coefficient | 0.313 ** | 1.000 |
| | | Sig. (2-tailed) | 0.000 | . |
| | | N | 162 | 162 |

** Correlation is significant at the 0.01 level (2-tailed).

Their farms' economic impact is also viewed differently according to small farmers' occupation, which is directly correlated to their education level. Labourers (whose main economic profession is something other than farmers, such as plumbers, tractor drivers, car mechanics, and couriers) consider that the economic impact of their farms consists mainly of ensuring their families' subsistence (61.3%), while farmers declare the same thing but with an even higher percentage of over 84%. When the small farmers' main occupation requires a professional level of training/education, such as legislators and higher officials (public administration officials or lawyers) or others (engineers, economists, IT technicians, doctors), they identified tourism as a viable complementarity to the agricultural development of their lavender business more often (Figure 3). The higher the occupational standing, the more small farmers are interested in diversifying the economic impact of lavender farms towards other domains such as tourism.

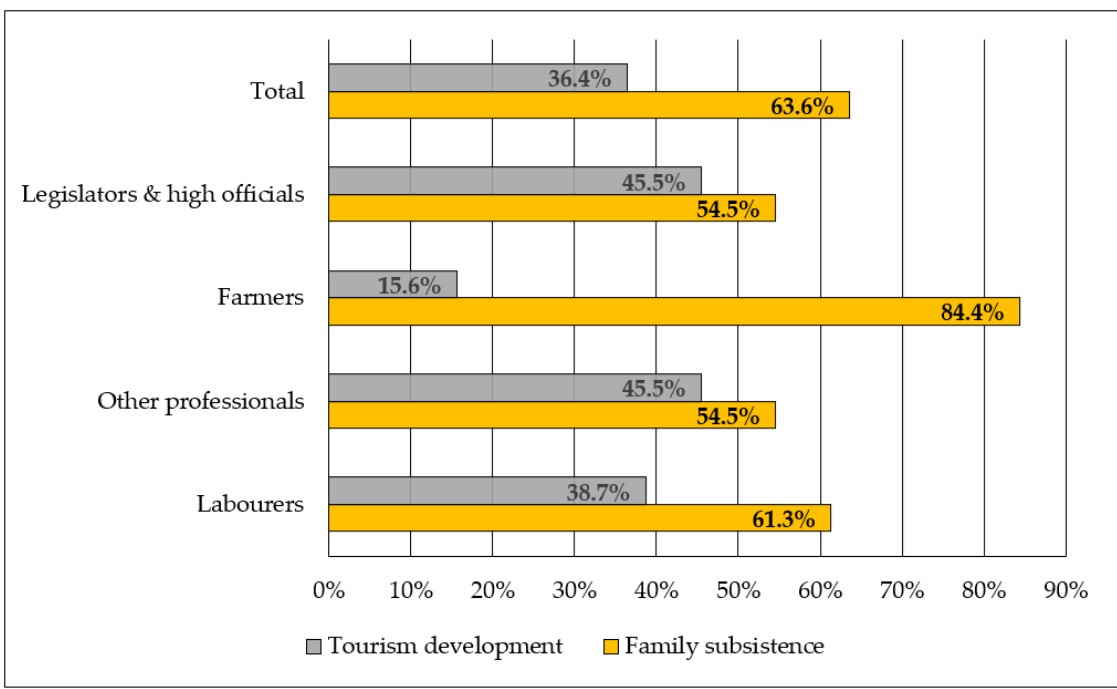

**Figure 3.** Correlation between lavender farms' economic impact and farmers' occupation.

Including tourism-related activities in their business stems from the following widely popular (especially among tourists) feature of lavender fields: their aesthetics [127,142]. In Romania, at this point, this phenomenon is just in its infancy, with small festivals (i.e., "open gates" events) organised inside lavender fields during their flowering period in June. *"We organise a small festival in June in Mădăraș, exactly in the fields; we have a 2 ha*

*plot, where we usually receive those who love the purple gold"* F, Mureș County—MR1; *"in June when the fragrance is so strong that it covers the entire 5 ha of the farm, we wait for people to come to our festival, so they can enjoy nature and relax in a lovely place; we took inspiration from what people are doing in Provence"* F, Brăila County—MR2). In addition to such events, farmers organise small brunches, picnics, or photo sessions, as follows: *"we organise picnics straight in the middle of the lavender fields, in Bonțida, where people can taste locally made lavender flavoured products—cookies, ice cream, lemonade"* F, Cluj County—MR1; *"for June we made unique decorations, we have a lavender farm in Fibiș where we organise professional photo sessions"* F, Timiș County—MR4. Interviews showed that these events have the added benefit of being an opportunity for small farmers to promote and commercialise their products, such as the following: essential oils, floral water, diffusers, creams, soaps, bath bombs, etc.

Small farmers with older lavender farms that have capitalised on their businesses by including them in other related domains, such as tourism, are also more satisfied with their development. The Spearman correlation coefficient shows a positive and statistically significant relationship (Sig. (2-tailed)—0.000) between duration (the length of time of their lavender farms/business) and satisfaction level. This strengthens the rationale that lavender farming can be a suitable solution for improving the economic and social environment of rural communities in Romania (Table 8).

**Table 8.** Spearman's rank correlation coefficient between duration and satisfaction level.

|                |              |                         | **Duration** | **Satisfaction** |
|----------------|--------------|-------------------------|--------------|------------------|
| **Spearman's rho** | Duration | Correlation Coefficient | 1.000        | 0.287 **         |
|                |              | Sig. (2-tailed)         | .            | 0.000            |
|                |              | N                       | 162          | 162              |
|                | Satisfaction | Correlation Coefficient | 0.287 **     | 1.000            |
|                |              | Sig. (2-tailed)         | 0.000        | .                |
|                |              | N                       | 162          | 162              |

** Correlation is significant at the 0.01 level (2-tailed).

Regardless of how old their lavender farm is (1, 2, 3, 4, 5, 5–10 years old), more than half of the interviewed farmers do their best to commercialise lavender products, not just the raw material. As they gain more experience and seniority in this business, the percentage of those who just cultivate lavender decreases and the diversity of ways in which the farms are capitalised increases. For example, they sell lavender cuttings alongside traditional products (handmade cosmetics, handmade decorative products, etc.). In total, 33.3% of those active for 4 years and 45.5% of those active between 5 and 10 years in the lavender business declared they engage in more than one type of commercial activity, which increases the farms' overall economic impact. The Chi-square test for the crosstab between duration and business development had a statistically significant $p$-value (Asimp. Sig. (2-sided)—0.027). Moreover, experienced farmers support new farmers with the knowledge they earned by ensuring free coaching to those who buy cuttings and want to start a lavender farm or by offering paid counselling to those who wish to start a business, as follows: *"I offer free counselling for the first 3 years to those to buy cuttings from me, but they can also contact me after that in case they face different problems"* F, Brăila County—MR2; *"when I deliver lavender cuttings, I also include a detailed technical chart with everything they need to know about cultivating lavender, about its planting and maintenance"* F, Timiș County—MR4).

Creating new jobs and new business trends are social elements that can improve a community's living conditions by increasing its income levels (and economic status). They are also elements that the small farmers identified as applicable to lavender farming. Respondents classified this social impact into different degrees of importance according to their education level and occupation (Table 9).

**Table 9.** Social impact according to education level and occupation.

| | | Social Impact | | Total |
|---|---|---|---|---|
| | | New Jobs | New Business Trend | |
| Profession | Labourers | 54.70% | 45.30% | 100.00% |
| | Other professionals | 56.80% | 43.20% | 100.00% |
| | Farmers | 71.90% | 28.10% | 100.00% |
| | Legislators and high officials | 45.50% | 54.50% | 100.00% |
| | TOTAL | 58.00% | 42.00% | 100.00% |
| Education level | Primary | 66.70% | 33.30% | 100.00% |
| | Highschool | 68.20% | 31.80% | 100.00% |
| | University or higher | 50.00% | 50.00% | 100.00% |
| | TOTAL | 58.00% | 42.00% | 100.00% |

Full-time farmers overwhelmingly (71%) indicated that the main social impact generated by lavender farming is the creation of new jobs. They mentioned a new business trend as a secondary effect and to a lesser degree than the other occupational categories.

Similar to the economic impact, the Spearman correlation coefficient shows a positive, but weaker, relationship between education level and social impact (Sig. (2-tailed)—0.023; correlation coefficient—0.178).

Identifying social opportunities correlates with the extent of their lavender business (i.e., duration). The hierarchical importance of the following two types of social impact differs: with start-up level farmers declaring that the most important social impact is creating a new business trend (62.5%), while farmers who have been active for more than 2 years place the creation of new jobs first (Figure 4).

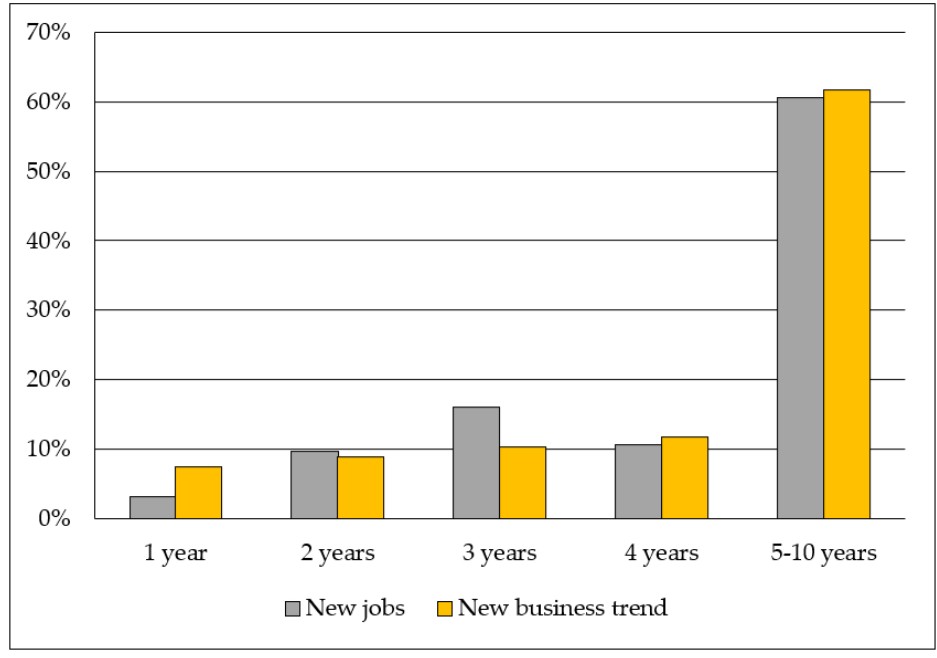

**Figure 4.** Relationship between the identified social impact of lavender farming and business duration.

Creating new jobs translates in reality into involving family members in the agricultural activities for farms younger than 2 years (*"we started to grow lavender two years ago, so far we are doing everything ourselves, family members trim the crops, take out the weeds, gather the flowers"* F, Teleorman County—MR3), and creating new seasonal or permanent jobs for community members for farms older than 4 years (*"our business is varied: we cultivate and maintain the fields, process the raw product, and commercialise various things from cuttings to*

*other finite lavender products, so we can no longer manage to do this with only family members; we need day labourers for maintenance works and harvesting, for rapidly processing the lavender flowers to send to distilleries, but also permanent employees for product packaging and labelling, online sales, marketing, promotion, and organising our events (festivals)"* F, Alba County—MR1; *"we need day labourers for harvesting and people to work in bottling esential oils but also people in sales and marketing; we need young people they are the best suited to deal with this online part of things"* F, Vrancea County—MR2).

The crosstabulation between the existence of a social or economic impact (family subsistence or tourism development; creating new jobs or new trends) and the envisaged length of their business presents a large number of responses (approx. 60%) that declare lavender farming to be a long term business (more than 10 years) (*"I think I can do this for a long time, I started growing lavender because I found out the plant has a long life and it becomes more profitable over time; it starts producing after 2 or 3 years depending on what kind of plant it is"* F, Botoșani County, MR2; *"once I invest in this I was told I can be sure I will have a secure income for 15–18 or even 20 years"* F, Alba County—MR1). This is to be expected given that lavender can flower from 15/18 to 20 years, depending on plant variety [31,50,51].

All the results confirm the research hypothesis, namely, that farming in Romania is dominated by the need to maintain the subsistence of the farmers' households, and as such, its social and economic impact is low. Nonetheless, our research shows a large diversity of the small farmers who take up lavender farming in Romania (i.e., education level or occupational profile). This influences how lavender businesses develop and their future social and economic impact.

*3.3. Ecological Impact of Small-Scale Lavender Farming*

According to the analysis of the interviews, over 94.5% of the interviewed small farmers mentioned that lavender farms are beneficial to their ecological systems. When asked to detail lavender's positive ecological effects, many mentioned their importance for beekeeping and maintaining or reintroducing biodiversity in the farms' areas. Some respondents referred directly to the species they have personally identified in their lavender fields (bees, bumblebees, butterflies, ladybugs, birds, rabbits, etc.). *"Lavender farms are a significant benefit for the environment, I think about the multitude of bees I see swarming in June when the flowers appear"* F, Brașov County—MR1; *"I do not need to use insecticides or pesticides in the lavender fields, so they are filled with bees and butterflies"* F, Prahova County—MR3; *"lavender benefits the bees, this field I have created a microsystem inside the agricultural landscape, like a shelter for small wild animals"* F, Olt County—MR4; *"the lavender fields can host many wild animals that can burrow safely (rabbits, foxes), there are no heavy machineries involved, no phonic pollution, and no emissions"* F, Timiș County—MR4; *"it benefits the environment, it is a medicinal plant, and it has a strong colour when it flowers so it attracts bees, butterflies, ladybugs"* F, Giurgiu County—MR3; *"as a medicinal and ornamental plant it is friendly with its environment, plus it is useful for bees, whenever we hear the bees buzzing we know the first flowers are blooming"*—F, Botoșani County, MR2.

Among these crops' other benefits were their value as natural insecticides and their aesthetic value, as follows: *"a beautiful, charming, inviting and decorative scenery"* F, Cluj County—MR1; *"a landscape with a purple field"* F, Vaslui County—MR 2; *"it beautifies the agricultural landscape"* F, Olt County—MR4; *"landscape with a nice visual aspect"* F, Satu Mare County—MR1; *"a splash of colour"* F, Alba County, MR1; *"a corner to relax in"* F, Galați County—MR2); *"an opportunity for aromatherapy straight in the middle of nature"* F, Brașov County—MR1; *"natural insecticide, we do not get any mosquitos"* F, Constanța County MR2; *"lavender helps the environment, us and the crops neighbouring the fields, they do not get any pests either"* F, Ialomița County—MR3.

The crosstabulation between perceived environmental impact and education level shows that 44% of those with a high school level education and 52.7% of those with a university degree or higher named multiple positive ecological effects of lavender farming. Its support for beekeeping and biodiversity was named more often by small farmers with a

university degree or higher (61%) than by those with a high school level education (34.1%). The first category of respondents also included the possibility of economic diversity as a benefit.

Regardless of duration, more than half of each category (from 1 year up to those who have had lavender farms for 10 years) observed the positive effects that lavender farming has on the environment. Similarly, the most often mentioned benefits were beekeeping and biodiversity (between 20 and 37% for the following categories: 1-, 2-, 3-, 4-, 5–10-year-old lavender farms).

Small farmers started their lavender business first and foremost as an economic venture, but the beneficial ecological consequences were not lost on them. Some of them declared that they had to choose between lavender farming and abandoning a plot that was degrading or investing labour and financial capital in it but cultivating the following more traditional but less profitable crops: *"lavender crops benefit the environment plus we are recapitalising on plots that would otherwise be abandoned"* F, Olt County—MR4; *"it was a good choice because we cultivated a plot that did not have anything growing on it a long time before we started our farm"* F, Vrancea County—MR2; *"we started lavender farming on an abandoned plot, we received it from an old lady, but we had to clean it first it was filled with thorns and weeds"* F, Satu Mare County—MR1.

Given that the civil servants with an agricultural degree have a higher education level and more extensive knowledge of the topic, they were also asked about the ecological effects of lavender farming. Similar to the small farmers interviewed, they identified this crop as a beneficial one, with most answers mentioning beekeeping and biodiversity (*"it is a plant with great melliferous properties"* CsA, Cluj County—MR1; *"this type of crop supports the local biodiversity"* CsA, Brăila County—MR2), but mentioned other aspects as well (*"lavender is a natural insecticide, as it repels pests"* CsA, Dolj County—MR3). One out of ten respondents mentioned the aesthetic landscape it creates (*"a plant that offers a visually pleasing landscape"* CsA, Botoșani County—MR2). Some respondents noted that lavender fixes the soil because of its vigorous and deep roots and thus has an anti-erosion role (*"their roots are very robust and it can stabilise soils prone to erosion"* CsA, Mureș County—MR1).

The relationship between the environmental impact of lavender farming and the perception of civil servants with agricultural expertise regarding the future of these types of businesses varies. However, the more optimistic their views, the more often we can see mentions regarding positive impacts such as supporting beekeeping and biodiversity (Table 10).

**Table 10.** Crosstabulation between the perception of civil servants with agricultural expertise regarding the future evolution of lavender farming and their positive environmental impact.

| | | | Positive Environmental Impact | | | | Total |
|---|---|---|---|---|---|---|---|
| | | | Beekeeping and Biodiversity | Natural Insecticide | Others | Aesthetic Landscape | |
| Perception future evolution | Sustainable | % within Perception Future Evolution | 50.0% | 9.1% | 27.3% | 13.6% | 100.0% |
| | | % of Total | 23.4% | 4.3% | 12.8% | 6.4% | 46.8% |
| | Growing business | % within Perception Future Evolution | 29.4% | 58.8% | 11.8% | - | 100.0% |
| | | % of Total | 10.6% | 21.3% | 4.3% | - | 36.2% |
| | Insecure business | % within Perception Future Evolution | 16.7% | 50.0% | - | 33.3% | 100.0% |
| | | % of Total | 2.1% | 6.4% | - | 4.3% | 12.8% |
| | NA | % within Perception Future Evolution | 50.0% | 50.0% | - | - | 100.0% |
| | | % of Total | 2.1% | 2.1% | - | - | 4.3% |
| Total | | % within Perception Future Evolution | 38.3% | 34.0% | 17.0% | 10.6% | 100.0% |
| | | % of Total | 38.3% | 34.0% | 17.0% | 10.6% | 100.0% |

In total, 70.21% of civil servants with agricultural expertise declared there is no negative ecological impact of lavender farming, and 29.79% said they did not know of any negative ecological effects *("in our region, no negative effects are known, on the contrary, the positive effects are evident"* CsA, Arad County), obviously validating this hypothesis.

## 4. Discussions

While the first hypothesis was validated and proved that those who participated in information and training programmes had a higher rate of applying for and accessing structural funds than those who had not, a few problems are worth discussing from our point of view.

The first one is the overall low rate of attendance. The degree to which small farmers participate in training and information programmes regarding structural funds is low. In total, 70% of respondents declared that they had not attended any training programme organised by CAD or APIA (*"because we are wasting our time, it is very difficult for us small farmers to access European funds given that we do not meet the eligibility criteria"* F, Vaslui County, MR2; *"I think attending these events can't help me access European money, I need a firm to write the project for me"* F, Bihor County, MR1). Some respondents also questioned the effectiveness of these programmes as follows: *"I didn't manage to submit any project"* (F, Tulcea County—MR 2); *"after submitting, they told me that I had a low score"* (F, Bihor County—MR 1). To bridge these shortcomings, lavender farmers mentioned the following: *"the need for an increased awareness of the administrative staff in managing projects"* (F, Vaslui County—MR2) to increase applications' success rate and avoid projects being denied financing.

Both target groups consider that the most needed action relates to accessing funds. Half of the small farmers interviewed requested reducing bureaucracy and abiding by the financing terms. Others, mainly lavender farm owners with high school level education, invoke the necessity of a more diversified financial mechanism, as follows: *"we should access European funds more easily; right now, there is too much bureaucracy. We need more help from the state or the Ministry of Agriculture so we can buy lavender planting and harvesting equipment. I had to get a bank loan to buy my harvester; applying for funds was too complicated"* (F, Teleorman County—MR3); *"we need specialists to help people apply for funds"* (F, Sălaj County—MR1); *"small farmers also need to be helped, even if they do not have an active business"* (F, Cluj County—MR1); *"we need to access funds more easily to buy special equipment"* (F, Dolj County—MR4). Similar comments were noted from farmers in Vaslui County—MR2.

The members of MADR agencies also drew attention to the large number of documents required (*"the procedures need simplifying, right now the dossier must contain up to 1200 pages"* CsA, Suceava County—MR2) and to the process' current low viability degree, proposing *"more advantageous programmes that target the entire chain from farming to processing to sales"* (CsA, Prahova County—MR3).

While PNDR is a general instrument to implement CAP in Romania, through which lavender farmers could also access financial grants, it proved restrictive due to their reduced entrepreneurial power (Table 2). Its limiting characteristics may have come from the modest efforts in promoting it, strict eligibility criteria relating to age (younger than 40 years old), or the requirement that applicants have graduated from agricultural-related educational institutions. The sample of farmers participating in this study had a median age of 43, and only 19.8% had domain-related qualifications. Not surprisingly, both target groups considered that future actions should include the following: the increase and diversification of financing mechanisms, accounting for the characteristics of the Romanian agricultural reality (i.e., where subsistence and semi-subsistence farms predominate), more transparent grant access procedures, and a smaller volume of documents needed for applying.

Both in this study and others analysing the same topic, small farmers and civil servants with agricultural expertise mentioned and recognised the need for better cooperation in order to revitalise the Romanian agricultural sector, support and encourage entrepreneurship in order to make it more profitable, all based on much simpler procedures for accessing the financing mechanisms and information dissemination actions [67].

Accessing the European agricultural funds has played and continues to play a leading role in the profitable development of agricultural holdings [143]. However, the overall absorption of European agricultural funds is low because the proposed projects are not fully adapted to the specificities of each EU member state [144]. There is still a solid need to finance and subsidise agricultural activities to ensure an inclusive rural development that simultaneously protects the environment and small farmers [145].

The training programmes for accessing European funds are an extremely helpful tool to increase the absorption rate of EU financing. However, the ability to organise and the degree of implementation in rural areas differ from one EU country to another, and, in many cases, the degree of participation is low. Many farmers, even young ones, use specialised consultancy firms to write their financing programmes or for advisory support and fund management to guarantee their success [146].

One way to improve this situation and increase the rate of small farmers obtaining structural funds should be to improve the governmental representatives' support during the application process, which can be challenging depending on the applicant's level of education, occupation, and digital skills. Another helpful action to improve the attendance rate would be to better promote the training programmes. The farmers' participation in these programmes would improve their ability to access structural funds and help finance their business, given that many lavender businesses in Romania are currently in their infancy. The start-up feature is explained by small farmers abandoning traditional crops and instead focusing on potentially more profitable crops. A powerful reason that inspires and encourages farmers to try and access structural funds or simply invest their own savings in lavender businesses in our country today is not necessarily the institutional framework that stimulates financing through structural funds. As the study of this hypothesis has proven, there are serious issues with organising, promoting, and implementing training programmes on this topic. On the contrary, what governs farmers' decisions is the successful model of other farms in the country. At this moment, we can even declare that the development of lavender businesses is an economic trend. In addition to the institutional efforts to implement CAP measures, the development of farmer coaching programmes aimed at increasing the economic productivity and sustainability of these farms [147] adapted to also include accessing structural funding or other know-how transfer systems could be another interesting area to consider and the starting point for future research for small-scale lavender farms in Romania.

The authors want to emphasise a series of issues regarding the results obtained while validating the second hypothesis. Currently, in Romania, the development of lavender farms is the result of the initiative of landowners whose properties fall under the definition of small or very small farms. These farmers primarily practise subsistence agriculture because they do not have alternative entrepreneurial perspectives that could offer them incomes higher than the usual seasonal crops. Many of these landowners do not gain access to the regional or even micro-regional supply chains for the agri-food systems that typically function in Europe and represent, according to researchers [96], 'part-time farms' or 'peasant farms.' Small farms' conversion to niche crops is efficient because the profits they could obtain are perceived to be higher than the sums invested in the enterprise, thus allowing farmers to reinvest their savings or obtain a bank loan to enlarge their business. Several studies from Turkey, India, the Kashmir region, and Spain have also concluded that lavender farming and related businesses can produce substantial incomes [37,40,42,148].

Recognised as one of the main factors for rural poverty alleviation in the future [106], agriculture, and particularly profitable niche crops such as lavender represent a suitable solution for Romania on the condition that production finds adequate and effective channels of distribution to dedicated markets. For certain countries, 'intermediary powers' in Albania [45] or farming associations in the Republic of Moldova [149,150] represent solutions for small farmers' businesses, allowing them to enter and compete in export markets. Both the quality of the crops and oil [33,38], as well as marketing availability [37], represent important elements to strengthen in order for the economic and social impact

of lavender farming in Romania to try and follow in the footsteps of similar small-scale holdings in countries with a tradition of lavender farming.

Lavender brings considerable opportunities, which government institutions have observed in a series of recent publications aimed at entrepreneurs who target this domain specifically (e.g., *The good practices guide for growing and harvesting medicinal and aromatic plants*, published by the Ministry of Agricultural and Rural Development) [151]. Even so, the Romanian strategic policy framework does not have any concrete measures meant to support lavender farming, as is the case in other states or regions where this crop has a traditionally significant socio-economic role (e.g., Fragrant Alps—France) [152].

Sometimes, the two target groups have different visions regarding the necessary actions and measures to improve lavender farming in Romania. For example, small farmers propose that the state intervene and create a functional, dedicated market for lavender products and implement better information dissemination channels. At the same time, the interviewed civil servants with agricultural expertise consider that this domain needs improved legislation, consultation of all stakeholders before redacting the Guides for Accessing Structural Funds (SF), but also intensified research support, as follows: *"there needs to be an improved orientation towards marketing and commercialising the products, increasing competitiveness, and also more strongly emphasising research in this domain, introducing and acquiring modern technologies, as well as enabling and increasing access to digitalisation"*, (CsA, Mehedinți County—MR4).

More than half of the interviewed civil servants with agricultural expertise declared that lavender farming is a somewhat or mostly successful business. However, when asked about the future development of lavender farming in Romania, 27.3% of those categorising it as presently being mostly successful and 20% of those seeing it as being somewhat successful declared that, in the long run, this is an insecure business. They have also detailed the main barriers that lavender farming faces nowadays, which may persist and/or amplify in the future. Those who currently view lavender farming as very or mostly successful identified an obvious obstacle in the lack of a dedicated market (*"the lack of a market niche for lavender products for small producers is very difficult to overcome"*, CsA, Teleorman County—MR3; *"there is no dedicated market for small lavender farmers because they produce small quantities as it is a new type of economic activity in Romania"* CsA, Iași County—MR2). Many of those who currently view lavender farming as mostly or very unsuccessful also mentioned other barriers, such as the lack of labour (27.3% and 16.7%, respectively) *"farmers who need workers to grow, process and commercialise lavender and/or adjacent products find it difficult because many people from rural areas migrated to Western Europe to look for better-paying jobs"* CsA, Prahova County—MR3; *"there is a lack of labour force because local people who already receive social aid or welfare are not interested in manual work, sometimes paying even less they already receive"* CsA, Vaslui County—MR2). An additional barrier is, in their opinion, the lack of association (*"association initiatives would prove more successful in organising and capitalising the final products"* CsA, Cluj County—MR1). Depending on what barriers they have identified, civil servants with agricultural expertise have different opinions about the future of the lavender business in Romania.

In Romania, small farmers and government representatives who have expertise in this field see lavender farming as a suitable solution precisely due to the specificity of Romanian agriculture, which translates into fragmented land plots with numerous subsistence or semi-subsistence farms that currently have poor or no economic efficiency. However, these small farms face grave issues that they will probably not overcome without government support. They have the potential to create new jobs, even more so, create local jobs due to the seasonality of some agricultural work, and permanent jobs in more varied positions (i.e., targeting young people and adults of all education levels), but there is no labour force pool from which to draw them. The lack of a labour force is caused by increased rural–urban outmigration and a general disinterest in the kinds of jobs and offered wages, with many people choosing to cash in on unemployment benefits or social aid. The issues of needing to increase competitiveness and profitability of the small agricultural holdings and labour

force scarcity could be counteracted with increased advanced mechanisation as well as ensuring the commercialisation of a more considerable number of lavender-based products. Another problem is the urgent need to acquire specialised tools and machines for lavender maintenance and processing, combined with difficulties and delays in accessing the PNDR financing mechanisms. Last but not least, the country does not have a well-developed dedicated market for lavender products, and their commercialisation is still incredibly difficult and slow.

Thus, while the currently measured social and economic impacts are low, lavender farming is a sector suitable to thrive in the future and solve some of the problems Romanian rural communities are currently facing.

In the analysis of the third hypothesis, all respondents from both target groups acknowledged that lavender farming has beneficial effects on the environment. Many of them offered concrete examples (honey-related properties, supporting the local biodiversity, natural insecticide, reduction of soil erosion, and ornamental effect). The same conclusions were identified by many researchers from the specialised international literature [28,127,133,134,136–140,148].

Due to current climate changes, lavender farming in Mediterranean areas that were traditionally known for this crop (i.e., Provence) is threatened by floods, droughts, excessive heat waves, and pest infestations, leaving farmers with only the option of destroying the affected crops [140,153–156]. Aridisation, which has much more profound effects in the Mediterranean region, has made Romania's territory favourable for extending lavender farming into Eastern Europe.

The interviews showed that none of the respondents from the two target groups provided any examples or arguments regarding the negative impact on the environment of lavender crops. The almost unanimous proportion of both groups who repeated the positive ecological effects of lavender on the environment clearly confirms the third hypothesis of the study.

The study presented certain limitations. Firstly, according to the research focus, our applied methodology excluded, on purpose, larger business actors. Despite their low number, they concentrate the largest share of the internal lavender oil market. However, they represent a distinctive business sector and face challenges different from small-scale farms. Their experience and perspective on this relatively new agriculture sector in Romania may be valuable input for follow-up research in the future.

Secondly, our study was primarily qualitative and did not allow for a precise, elaborate statistical analysis, but instead was guided by the specialised literature that encourages the use of mixed methods.

Thirdly, the pandemic-related regulations imposed a limitation during the first phase of interviews (December 2019 and March 2020). Because of the lockdown periods and the sanitary context, particularly critical in Romania, we switched from a face-to-face perspective to phone interviews and encountered difficulties imposed by some respondents' lack of digital skills. This was why we enlarged our sample size through the second phase of interviews in March 2022, when the pandemic-related sanctions were relaxed and allowed for events where we could meet the small farmers face-to-face.

Lastly, we faced a statistical issue in that the total number of small and very small lavender farms in Romania does not appear in the official statistical reports. In reality, the available data reflect the number of farms that meet specific criteria and whose owners have taken steps to request financial subsidies as per APIA requirements, as mentioned in Section 1.1.3. EU Agricultural Policies and Section 2.2. Methodological Tools and Sample Size.

## 5. Conclusions

This research represents an exploratory perspective that may help both scientists and professionals design improved studies for this agricultural sector because lavender offers real opportunities for small-scale farming that in Romania often displays subsistence char-

acteristics. Furthermore, decision-makers and politicians may also find it an informative source when making decisions and shaping policies to formulate appropriate economic and environmental effective measures for small agricultural businesses.

Romanian agriculture is mainly characterised by a fractured landscape of small plots, where subsistence and semi-subsistence farms predominate, marked by a low economic efficiency. In this context, small farmers (owners of these subsistence and semi-subsistence farms) and governmental representatives see lavender farming as a durable and successful solution that can bring positive changes to rural communities.

Our first finding was that small farmers are not very interested in information and training programmes meant to help them access structural funds and increase the absorption rate of European funding. Most of the small farmers participating in this research started their business using primarily their savings, followed by those who added bank loans as a complementary source to their private savings and the smaller category of respondents who also accessed structural funds.

Our second finding was that lavender farms' social and economic impact is low at a national level, but they are important at a community level. The obvious economic benefits for small farmers relate to the financial aspect for the entrepreneurial families and the possibility to develop other activities such as tourism. The social benefits that derive from the economic ones, as shown by this research, are that lavender farming can help increase the living standards of rural communities by creating new seasonal or permanent jobs in agriculture, processing, and/or commercialising lavender and lavender-based products. At the same time, this type of crop is a new business trend among Romanian farmers and can boost the entrepreneurial sector of the country. The research showed that the success rate of lavender farms is higher when the farm is older (i.e., its duration is longer) (e.g., 5–10 years), so for many currently start-up farms, there are numerous possibilities for entrepreneurial initiatives (e.g., tourism). The research also identified the following obstacles faced by small lavender farmers that slow down the development of lavender farming in Romania: the weak outlet market or the lack of a dedicated market, the lack of a labour force, the weak associative power of these small and very small farms, difficulties in accessing structural funds, the necessity to buy specialised tools or machines for maintaining the crops and for processing the lavender.

Thirdly, the research found that the impact that lavender farming has on the environment is mainly positive. The benefits of this crop were identified by the respondents of both target groups and support the durability of this crop, all exceeding any possible negative effect. The environmental benefits of the crop include supporting beekeeping and biodiversity, its natural insecticide feature, and the aesthetic value of this crop, which also supports tourism.

Considering all the relevant data shown by this study, the theme of small farm development in a competitive and performing agricultural context still requires investigation, and the authors will continue their research to identify solutions for harmonising the Romanian specifics of agricultural reality with the Common Agricultural Policy.

Future research could focus on a more in-depth market analysis of this niche product that presents real economic opportunities for a sector with a very inconsistent income for small household farms in Romania. A multidimensional perspective involving a canonical analysis and a complementary quantitative study on lavender farming in Romania could be useful to depict a better image of the present market for this product and surpass its possible evolutionary trends.

A valuable research topic for future studies would be a comparison perspective among internal regions and between Romania and emerging neighbouring producers (e.g., the Republic of Moldova), which also represent competitors in the lavender oil international market and may offer good practice examples.

**Supplementary Materials:** The following supporting information can be downloaded at: https://www.mdpi.com/article/10.3390/land11050662/s1, Interview Guide S1: Interview guide regarding lavender farming, targeting the owners of small farms in Romania; Interview Guide S2: Interview guide regarding lavender farming, targeting the civil servants with agricultural expertise in Romania.

**Author Contributions:** The authors contributed equally to this work. Conceptualization, I.V., E.M., and A.-I.L.-D.; methodology, A.-I.L.-D. and E.M.; software, A.-I.L.-D.; validation, A.-I.L.-D., I.V., and M.P.; formal analysis, A.-I.L.-D. and M.P.; investigation, A.M.; resources, A.-I.L.-D., I.V.; data curation, M.P.; writing—original draft preparation, I.V., A.-I.L.-D., and A.M.; writing—review and editing, I.V., A.M., and A.-I.L.-D.; visualization, A.-I.L.-D. and M.P.; supervision, I.V. and A.M.; project administration, I.V.; funding acquisition, I.V. All authors have read and agreed to the published version of the manuscript.

**Funding:** The publication of this research has been partially funded by the University of Bucharest, Romania.

**Acknowledgments:** The authors would like to thank Roxana Cuculici (Department of Regional Geography and Environment, University of Bucharest, Romania) for her valuable input and recommendations in constructing the cartographic material of this article. The authors would also like to thank the editors and anonymous reviewers for their constructive comments and helpful suggestions.

**Conflicts of Interest:** The authors declare no conflict of interest.

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
