# Peer review of "Could Lavender Farming Go from a Niche Crop to a Suitable Solution for Romanian Small Farms?"

_land, doi:10.3390/land11050662_

Round 1
Reviewer 1 Report
I accept the explanations and appraciate the changes. I think the paper is much better in its present form.
Author Response
Dear reviewer,
We take this opportunity to thank you for accepting to analyse our paper and offer constructive suggestions that would improve it.
Sincerely yours,
The authors
Reviewer 2 Report
After two rounds of revisions, I am happy to recommend this paper for publication in Land without requiring additional revisions.
Author Response

(The authors gave the same response as above.)

Reviewer 3 Report
Dear Authors,
With a number of changes, I consider the manuscript to be of much better quality. However, I propose to justify the research hypotheses in a better way. H1 in particular requires a better substantive justification, not just an intuitive one.
In the introduction, I propose to describe the research gap more clearly and add the layout of the manuscript.
I also believe that an interview questionnaire should be available.
Author Response
Dear reviewer,
We take this opportunity to thank you for accepting to analyse our paper and offer constructive suggestions that would improve it.
Sincerely yours,
The authors

This manuscript is a resubmission of an earlier submission. The following is a list of the peer review reports and author responses from that submission.
Round 1
Reviewer 1 Report
Although the article is valuable, I have a few comments. I think making the following corrections will raise the scientific value of the article.
Comments:
- In the first section of the article it is worth considering discussing more detailed results of similar studies by other authors. The Discussion section refers in a very synthetic way to the results of studies conducted in other countries (e.g. Turkey, India). It might be worth considering a more detailed reference to other studies in this section as well.
- It is worth explaining why weights were not given to diagnostic variables,
- Please explain how the size of the research sample was determined (especially for small farmers). Was a minimum sample size formula used (what error rate was assumed, what level of significance, etc.)?
- Given the multidimensionality of the categories considered, it is worth considering the use of canonical analysis in the future (which could be mentioned in the conclusion),
- In the last paragraph of the summary, it is worth describing in more detail what the authors' further research will concern (e.g. what methods, what territorial area, whether the research will be international),
- Please add a few limitations sentences. What, according to the authors, may be a limitation of such research, which is based on primary sources.
Good luck,
Author Response
Dear reviewer,
We take this opportunity to thank you for accepting to analyse our paper and offer constructive suggestions that would improve it. While we admit that we are limited by the time constraints and our experience and expertise level in fully and satisfactory answering all your observations, please be assured that we have made all posible efforts to respond and that we will use the experience gained during this peer-review stage to improve our future endeavours.
We feel fortunate to have received your comments and we hope that we improved the manuscript well enough to be accepted for publication.
Sincerely yours,
The authors

Reviewer 2 Report
This is a very descriptive paper dealing with a quite marginal issue of lavender farming in Romania.
The paper is based on rather small surveys: 65 farm owners and 31 civil servants of people related to lavender farming. The technique used for analysis was a rather simple statistics with an extensive graphical representation.
The paper essentially summarizes the results of the small survey without creating a paper of higher general interest.
A few minor comments.
Line 13: I would recommend remove “using SPSS v.28” from abstract – it is not usual to include such references to statistical packages in article abstract.
Line 46 In Commune Agricultural Policy I would recommend to use Common instead of Commune.
Lines 81-84: It is not clear which farms are located where and what is the main lavender region.
Page 3: Focus should be on lavender farming, not on farming in general.
Line 203: semi-subsistance and subsistence farms should be briefly, but clearly, defined here.
Author Response

(The authors gave the same response as above.)

Reviewer 3 Report
Dear Authors,
Thank you for the opportunity to read the interesting research results. In my opinion, some issues should be made clearer:
1. Introduction: no clear presentation of the research gap, what are the goals of the research. Why was this hypothesis made? What is its substantive justification?
2. What do the authors understand by the "sustainable solution"? The concept shows up in the title, the hypothesis, but it is not clear how it is tested.
3. How was the hypothesis verified? What were the methodology and tests used?
4. There is a missing reference to the hypothesis in the results of the research and in the conclusion.
5. I propose to expand the Conclusion section with limitations, implications for farmers, rulers, residents, and future research
Minor:
Avoid so many references in one sentence. (line 34 - too many references in one sentence)
Author Response

(The authors gave the same response as above.)

Round 2
Reviewer 2 Report
The authors improved the quality of the paper by incorporating the reactions to my minor comments.
While the revised paper is a better one than the originally submitted paper, my major comments about too small sample and rather elementary level of analysis still apply. Therefore I still do not recommend the paper for publication.
Reviewer 3 Report
The authors considered all my comments. However, removing the question of the hypothesis significantly weakened the substantive level of the manuscript. The research is on a small sample, and introducing hypotheses would significantly increase the quality of research and results.